# A Survey on Transformers in Reinforcement Learning

**Wenzhe Li**[1*]                                                 *lwz21@mails.tsinghua.edu.cn*
**Hao Luo**[2,3*]                                                          *lh2000@pku.edu.cn*
**Zichuan Lin**[4*]                                                  *zichuanlin@tencent.com*
**Chongjie Zhang**[5†]                                                  *chongjie@wustl.edu*
**Zongqing Lu**[2,3†]                                                *zongqing.lu@pku.edu.cn*
**Deheng Ye**[4†]                                                      *dericye@tencent.com*

[1]*Tsinghua University*
[2]*Peking University*
[3]*BAAI*
[4]*Tencent Inc.*
[5]*Washington University in St.Louis*

**Reviewed on OpenReview:** *https://openreview.net/forum?id=r30yuDPvf2*

## Abstract

Transformer has been considered the dominating neural architecture in NLP and CV, mostly under supervised settings. Recently, a similar surge of using Transformers has appeared in the domain of reinforcement learning (RL), but it is faced with unique design choices and challenges brought by the nature of RL. However, the evolution of Transformers in RL has not yet been well unraveled. In this paper, we seek to systematically review motivations and progress on using Transformers in RL, provide a taxonomy on existing works, discuss each sub-field, and summarize future prospects.

## 1 Introduction

Reinforcement learning (RL) provides a mathematical formalism for sequential decision-making. By utilizing RL, we can acquire intelligent behaviors automatically. While RL has provided a general framework for learning-based control, deep neural networks, as a way of function approximation with high capacity, have been enabling significant progress along a wide range of domains (Silver et al., 2016; Vinyals et al., 2019; Ye et al., 2020a;b).

While the generality of deep reinforcement learning (DRL) has achieved tremendous developments in recent years, the issue of sample efficiency prevents its widespread use in real-world applications. To address this issue, an effective mechanism is to introduce inductive biases into the DRL framework. One important inductive bias in DRL is *the choice of function approximator architectures*, such as the parameterization of neural networks for DRL agents. However, compared to efforts on architectural designs in supervised learning (SL), how to design architectures for DRL has remained less explored. Most existing works on architectures for RL are motivated by the success of the (semi-) supervised learning community. For instance, a common practice to deal with high-dimensional image-based input in DRL is to introduce convolutional neural networks (CNN) (LeCun et al., 1998; Mnih et al., 2015); another common practice to deal with partial observability is to introduce recurrent neural networks (RNN) (Hochreiter & Schmidhuber, 1997; Hausknecht & Stone, 2015).

In recent years, the Transformer architecture (Vaswani et al., 2017) has revolutionized the learning paradigm across a wide range of SL tasks (Devlin et al., 2018; Dosovitskiy et al., 2020; Dong et al., 2018) and demonstrated superior performance over CNN and RNN. Among its notable benefits, the Transformer architecture enables modeling long dependencies and has excellent scalability (Khan et al., 2022). Inspired by the success of SL, there has been a surge of interest in applying Transformers in RL, with the hope of carrying the benefits of Transformers to the RL field.

---

* Equal contribution; † Equal advising.

The use of Transformers in RL dates back to Zambaldi et al. (2018), where the self-attention mechanism is used for relational reasoning over structured state representations. Afterward, many researchers seek to apply self-attention for representation learning to extract relations between entities for better policy learning (Vinyals et al., 2019; Baker et al., 2019). Besides leveraging Transformers for state representation learning, prior works also use Transformers to capture multi-step temporal dependencies to deal with the issue of partial observability (Parisotto et al., 2020; Parisotto & Salakhutdinov, 2021). More recently, offline RL (Levine et al., 2020) has attracted attention due to its capability to leverage large-scale offline datasets. Motivated by offline RL, recent efforts have shown that the Transformer architecture can directly serve as a model for sequential decisions (Chen et al., 2021; Janner et al., 2021) and generalize to multiple tasks and domains (Lee et al., 2022; Carroll et al., 2022).

The purpose of this survey is to present the field of *Transformers in Reinforcement Learning*, denoted as "Transformer-based RL". Although Transformer has been considered one of the most popular models in SL research at present (Devlin et al., 2018; Dosovitskiy et al., 2020; Bommasani et al., 2021; Lu et al., 2021), it remains to be less explored in the RL community. In fact, compared with the SL domain, using Transformers in RL as function approximators faces unique challenges. First, the training data of RL is collected by an ever-changing policy during optimization, which induces non-stationarity for learning a Transformer. Second, existing RL algorithms are often highly sensitive to design choices in the training process, including network architectures and capacity (Henderson et al., 2018). Third, Transformer-based architectures often suffer from high computational and memory costs, making it expensive in both training and inference during the RL learning process. For example, in the case of AI for video game-playing, the training performance is closely related to the efficiency of sample generation, which is restricted by the computational cost of the RL policy network and value network (Ye et al., 2020a; Berner et al., 2019). Fourthly, compared to models that rely on strong inductive biases, Transformer models typically need a much larger amount of training data to achieve decent performance, which usually exacerbates the sample efficiency problem of RL. Despite all these challenges, Transformers are becoming essential tools in RL due to their high expressiveness and capability. However, they are utilized for various purposes stemming from orthogonal advances in RL, such as **a) RL that requires strong representation or world model (e.g., RL with high-dimensional spaces and long horizon)**; **b) RL as a sequence modeling problem**; and **c) Pre-training large-scale foundation models for RL**. In this paper, we seek to provide a comprehensive overview of Transformer-based RL, including a taxonomy of current methods and the challenges. We also discuss future perspectives, as we believe the field of Transformer-based RL will play an important role in unleashing the potential impact of RL, and this survey could provide a starting point for those looking to leverage its potential.

We structure the paper as follows. Section 2 covers background on RL and Transformers, followed by a brief introduction on how these two are combined together. In Section 3, we describe the evolution of network architecture in RL and the challenges that prevent the Transformer architecture from being widely explored in RL for a long time. In Section 4, we provide a taxonomy of Transformers in RL and discuss representative existing methods. Finally, we summarize and point out potential future directions in Section 5.

## 2 Problem Scope

### 2.1 Reinforcement Learning

In general, Reinforcement Learning (RL) considers learning in a Markov Decision Process (MDP) $\mathcal{M} = \langle \mathcal{S}, \mathcal{A}, P, r, \gamma, \rho_0 \rangle$, where $\mathcal{S}$ and $\mathcal{A}$ denote the state space and action space respectively, $P(s'|s, a)$ is the transition dynamics, $r(s, a)$ is the reward function, $\gamma \in (0, 1)$ is the discount factor, and $\rho_0$ is the distribution of initial states. Typically, RL aims to learn a policy $\pi(a|s)$ to maximize the expected discounted return $J(\pi) = \mathbb{E}_{\pi, P, \rho_0} \left[ \sum_t \gamma^t r(s_t, a_t) \right]$. To solve an RL problem, we need to tackle two different parts: learning to represent states and learning to act. The first part can benefit from inductive biases (e.g., CNN for image-based states, and RNN for non-Markovian tasks). The second part can be solved via behavior cloning (BC), model-free or model-based RL. In the following part, we introduce several specific RL problems related to advances in Transformers in RL.

**Offline RL.** In offline RL (Levine et al., 2020), the agent cannot interact with the environment during training. Instead, it only has access to a static offline dataset $\mathcal{D} = \{(s, a, s', r)\}$ collected by arbitrary policies. Without exploration, modern offline RL approaches (Fujimoto et al., 2019; Kumar et al., 2020; Yu et al., 2021b) constrain the learned policy close to the data distribution, to avoid out-of-distribution actions that may lead to overestimation.

Recently, in parallel with typical value-based methods, one popular trend in offline RL is RL via Supervised Learning (RvS) (Emmons et al., 2021), which learns an outcome-conditioned policy to yield desired behavior via SL.

**Goal-conditioned RL.** Goal-conditioned RL (GCRL) extends the standard RL problem to goal-augmented setting, where the agent aims to learn a goal-conditioned policy $\pi(a|s, g)$ that can reach multiple goals. Prior works propose to use various techniques, such as hindsight relabeling (Andrychowicz et al., 2017), universal value function (Schaul et al., 2015), and self-imitation learning (Ghosh et al., 2019), to improve the generalization and sample efficiency of GCRL. GCRL is quite flexible as there are diverse choices of goals. We refer readers to (Liu et al., 2022) for a detailed discussion around this topic.

**Model-based RL.** In contrast to model-free RL which directly learns the policy and value functions, model-based RL learns an auxiliary dynamic model of the environment. Such a model can be directly used for planning (Schrittwieser et al., 2020), or generating imaginary trajectories to enlarge the training data for any model-free algorithm (Hafner et al., 2019). Learning a model is non-trivial, especially in large or partially observed environments where we first need to construct the representation of the state. Some recent methods propose to use latent dynamics (Hafner et al., 2019) or value models (Schrittwieser et al., 2020) to address these challenges and improve the sample efficiency of RL.

## 2.2 Transformers

Transformer (Vaswani et al., 2017) is one of the most effective and scalable neural networks to model sequential data. The key idea of Transformer is to incorporate *self-attention* mechanism, which could capture dependencies within long sequences in an efficient manner. Formally, given a sequential input with $n$ tokens $\left\{\mathbf{x}_i \in \mathbb{R}^d\right\}_{i=1}^n$, where $d$ is the embedding dimension, the self-attention layer maps each token $\mathbf{x}_i$ to a query $\mathbf{q}_i \in \mathbb{R}^{d_q}$, a key $\mathbf{k}_i \in \mathbb{R}^{d_k}$, and a value $\mathbf{v}_i \in \mathbb{R}^{d_v}$ via linear transformations, where $d_q = d_k$. Let the sequence of inputs, queries, keys, and values be $\mathbf{X} \in \mathbb{R}^{n \times d}, \mathbf{Q} \in \mathbb{R}^{n \times d_q}, \mathbf{K} \in \mathbb{R}^{n \times d_k}$, and $\mathbf{V} \in \mathbb{R}^{n \times d_v}$, respectively. The output of the self-attention layer $\mathbf{Z} \in \mathbb{R}^{n \times d_v}$ is a weighted sum of all values:

$$\mathbf{Z} = \text{softmax}\left(\frac{\mathbf{Q}\mathbf{K}^T}{\sqrt{d_q}}\right)\mathbf{V}.$$

With the self-attention mechanism (Bahdanau et al., 2014) as well as other techniques, such as multi-head attention and residual connection (He et al., 2016), Transformers can learn expressive representations and model long-term interactions.

Due to the strong representation capability and excellent scalability, the Transformer architecture has demonstrated superior performance over CNN and RNN across a wide range of supervised and unsupervised learning tasks. So a natural question is: can we use Transformers to tackle the problems (i.e., learning to represent states, and learning to act) in RL?

## 2.3 Combination of Transformers and RL

We notice that a growing number of works are seeking to combine Transformers and RL in diverse ways. In general, Transformers can be used as one component for RL algorithms, e.g., a representation module or a dynamic model. Transformers can also serve as one whole sequential decision-maker. Figure 1 provides a sketch of Transformers' different roles in the context of RL.

# 3 Network Architecture in RL

Before presenting the taxonomy of current methods in Transformer-based RL, we start by reviewing the early progress of network architecture design in RL, and summarize their challenges. We do this because Transformer itself is an advanced neural network and designing appropriate neural networks contributes to the success of DRL.

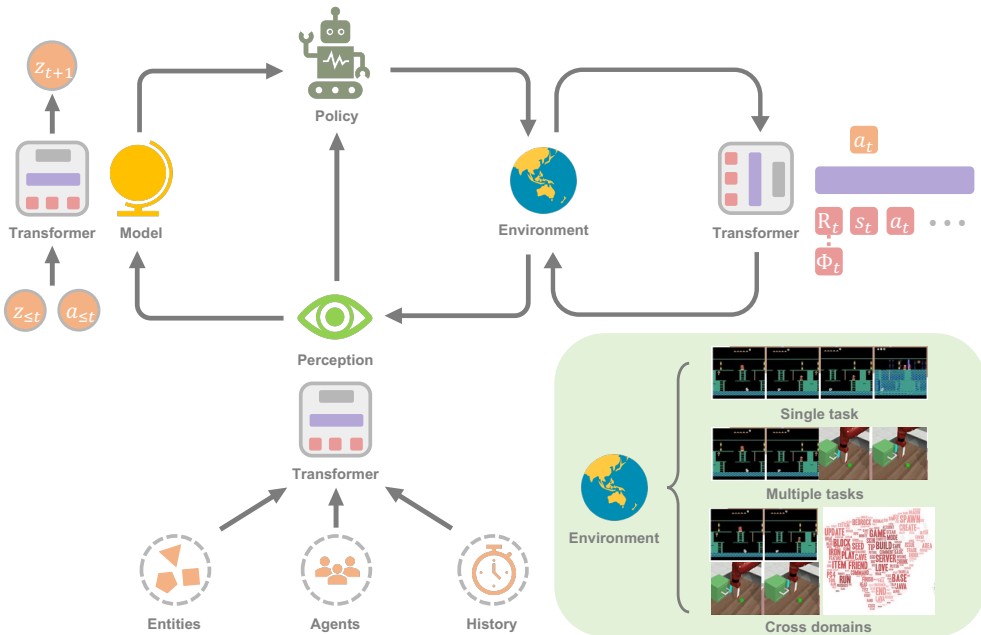

Figure 1: An illustrating example of Transformer-based RL. On the one hand, Transformers can be used as one component in RL. Particularly, Transformers can encode diverse sequences, such as entities, agents, and stacks of historical information; and it is also an expressive predictor for the dynamics model. On the other hand, Transformers can integrate all subroutines in RL and act as a sequential decision-maker. Overall, Transformers can improve RL's learning efficiency in single-task, multi-task, and cross-domain settings.

### 3.1 Architectures for function approximators

Since the seminal work Deep Q-Network (Mnih et al., 2015), many efforts have been made in developing network architectures for DRL agents. Improvements in network architectures in RL can be mainly categorized into three classes. The first class is to design a new structure that incorporates RL inductive biases to ease the difficulty of training policy or value functions. For example, Wang et al. (2016) propose dueling network architecture with one for the state value function and another for the state-dependent action advantage function. This choice of architecture incorporates the inductive bias that generalizes learning across actions. Other examples include the value decomposition network, which has been used to learn local Q-values for individual agent (Sunehag et al., 2017) or sub-reward (Lin et al., 2019). The second class is to investigate whether general techniques of neural networks (e.g., regularization, skip connection, batch normalization) can be applied to RL. To name a few, Ota et al. (2020) find that increasing input dimensionality while using an online feature extractor can boost state representation learning, and hence improve the performance and sample efficiency of DRL algorithms. Sinha et al. (2020) propose a deep dense architecture for DRL agents, using skip connections for efficient learning, with an inductive bias to mitigate data-processing inequality. Ota et al. (2021) use DenseNet (Huang et al., 2017) with decoupled representation learning to improve flows of information and gradients for large networks. The third class is to scale DRL agents for distributional learning. For instance, IMPALA (Espeholt et al., 2018) have developed distributed actor-learner architectures that can scale to thousands of machines without sacrificing data efficiency. Recently, due to the superior performance of Transformers, some researchers have attempted to apply Transformers architecture in policy optimization algorithms, but found that the vanilla Transformer design fails to achieve reasonable performance in RL tasks (Parisotto et al., 2020).

### 3.2 Challenges

While the usage of Transformers has made rapid progress in SL domains in past years, applying them in RL is not straightforward with the following unique challenges in two folds[1]:

---

[1]Namely Transformers in RL versus Transformers in SL, and Transformers in RL versus other neural network architectures in RL.

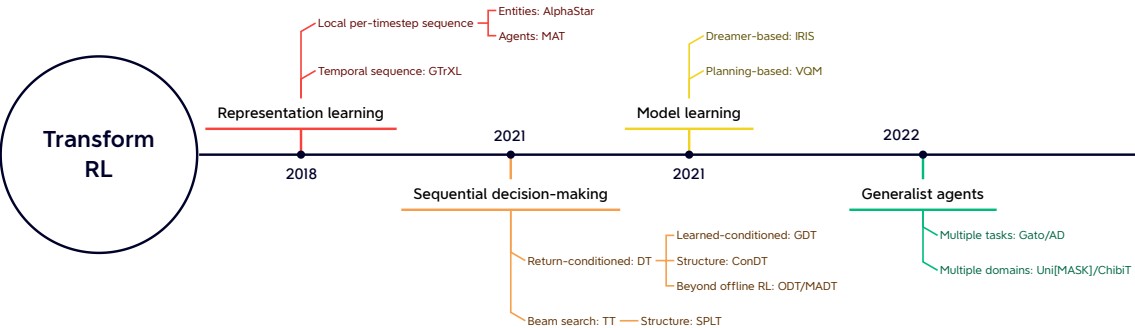

Figure 2: The taxonomy of Transformer-based RL (Transform RL in short). The timeline is based on the first work related to the branch.

On the one hand, from the view of RL, many researchers point out that existing RL algorithms are incredibly sensitive to architectures of deep neural networks (Henderson et al., 2018; Engstrom et al., 2019; Andrychowicz et al., 2020). First, the paradigm of alternating between data collection and policy optimization (i.e., data distribution shift) in RL induces non-stationarity during training. Second, RL algorithms are often highly sensitive to design choices in the training process. In particular, when coupled with bootstrapping and off-policy learning, learning with function approximators can diverge when the value estimates become unbounded (i.e., "deadly triad") (Van Hasselt et al., 2018). More recently, Emmons et al. (2021) identify that carefully choosing model architectures and regularization are crucial for the performance of offline DRL agents.

On the other hand, from the view of Transformers, the Transformer-based architectures suffer from large memory footprints and high latency, which hinder their efficient deployment and inference. Recently, many researchers aim to improve the computational and memory efficiency of the original Transformer (Tay et al., 2022), but most of these works focus on SL domains. In the context of RL, Parisotto & Salakhutdinov (2021) propose to distill learning progress from a large Transformer-based learner model to a small actor model to bypass the high inference latency of Transformers. However, these methods are still expensive in terms of memory and computation. So far, the idea of efficient or lightweight Transformers has not yet been fully explored in the RL community. In addition to considerable memory usage and high latency, Transformer models often require a significantly larger amount of training data to achieve comparable performance to models that rely on strong inductive biases (e.g., CNN and RNN). Given that DRL algorithms often struggle with sample efficiency, it can be challenging for DRL agents to gather sufficient data to train the Transformer models. As we shall see later, such a challenge inspires Transformer's prosperity in offline RL.

## 4    Transformers in RL

Although Transformer has become a popular model in most supervised learning research, it has not been widely used in the RL community for a long time due to the aforementioned challenges. Actually, most early attempts of Transformer-based RL apply Transformers for state representation learning or providing memory information, but still use standard RL algorithms such as temporal difference learning and policy optimization for agent learning. Therefore, these methods still suffer from challenges from the conventional RL framework. Until recently, offline RL allows learning optimal policy from large-scale offline data. Inspired by offline RL, recent works further treat the RL problem as a conditional sequence modeling problem on fixed experiences, which bypasses the challenges of bootstrapping error in traditional RL, allowing Transformer to unleash its powerful sequential modeling ability.

In this survey paper, we retrospect the advances of Transformer-based RL, and provide a taxonomy to present the current methods. We categorize existing methods into four classes: representation learning, model learning, sequential decision-making, and generalist agents. Figure 2 provides a taxonomy sketch with a subset of corresponding works.

## 4.1 Transformers for representation learning

One natural usage of Transformers is to use it as a sequence encoder. In fact, various sequences in RL tasks require processing[2], such as local per-timestep sequence (multi-entity sequence (Vinyals et al., 2019; Baker et al., 2019), multi-agent sequence (Wen et al., 2022) , etc.), temporal sequence (trajectory sequence (Parisotto et al., 2020; Banino et al., 2021)), and so on.

### 4.1.1 Encoder for local per-timestep sequence

The early notable success of this method is to process complex information from a variable number of entities scattered in the agent's observation with Transformers. Zambaldi et al. (2018) first propose to capture relational reasoning over structured observation with multi-head dot-product attention, which is subsequently used in AlphaStar (Vinyals et al., 2019) to process multi-entity observation in the challenging multi-agent game StarCraft II. The proposed entity Transformer encodes the observation as:

$$\text{Emb} = \text{Transformer}(e_1, \cdots, e_i, \cdots),$$

where $e_i$ represents the agent's observation on entity $i$, either directly sliced from the whole observation or given by an entity tokenizer.

Several follow-up works have enriched entity Transformer mechanisms. Hu et al. (2020) propose a compatible decoupling policy to explicitly associate actions to various entities and exploit an attention mechanism for policy explanation. Wang et al. (2023b) learn an entity Transformer with general knowledge and feature-space-agnostic tokenization via transfer learning within different types of games. To solve the challenging one-shot visual imitation, Dasari & Gupta (2021) use Transformers to learn a representation focusing on task-specific elements.

Similar to entities scattered in observation, some works exploit Transformers to process other local per-timestep sequences. Tang & Ha (2021) leverage the attention mechanism to process sensory sequence and construct a policy that is permutation invariant w.r.t. inputs. In the incompatible multi-task RL setting, Transformers are proposed to extract morphological domain knowledge (Kurin et al., 2020). Regarding the presence of multimodal information (e.g. image and language) in local per-timestep observations, Team et al. (2021) utilize Transformer-based structure to integrate these multimodal information and represent the state of the agent.

Moreover, recent RL algorithms are trying to incorporate vision inductive biases into policy learning. For instance, Vision Transformer (ViT) uses patch sequences to process images in the visual domain, which can be used for representation learning in RL. Tao et al. (2022) test the effectiveness of ViT and its variants combined with various self-supervised techniques (Data2vec, MAE, and Momentum Contrastive learning) for visual control tasks. However, no notable performance gain is shown in their experiments with less complex tasks. On the other hand, Hansen et al. (2021) find ViT-based architecture is prone to overfitting and address the problem with data-augmentation. Moreover, Kalantari et al. (2022) use ViT architecture to learn Q value with vision inputs, showing its potential to improve the sample efficiency of RL algorithms. Moreover, Seo et al. (2022a) combine ViT with an improved feature-mask MAE to learn image features that are better suited for dynamics, which can benefit decision-making and control.

As a summary, the Transformer, as a choice for local per-timestep sequence encoding, is being applied in various RL scenarios. When the RL task itself requires attention to the relationships among various parts of observations, such as entities and morphological domain sequences, the attention mechanism inherent in Transformers is suitable for their encoding process. When complex observation information needs to be processed, some works desire to transfer the expressive power demonstrated by Transformers in vision or language domains to the representation learning in RL. Additionally, there are some works indicating a trend towards using pre-trained encoders in RL, further establishing a deeper connection between the choice of representation learning structures in RL and the highly acclaimed architectures in the vision and language domains.

---

[2]In the context of Transformers, the term sequence refers to the manner in which data is processed. While the local per-timestep information may be represented as a set or graph structure in specific tasks, the information is conventionally aggregated in a specific order either as input or as a post-processing step in the output.

### 4.1.2 Encoder for temporal sequence

Meanwhile, it is also reasonable to process temporal sequence with Transformers. Such a temporal encoder works as a memory module:

$$\text{Emb}_{0:t} = \text{Transformer}(o_0, \cdots, o_t),$$

where $o_t$ represents the agent's observation at timestep $t$ and $\text{Emb}_{0:t}$ represents the embedding of historical observations from initial observation to current observation.

In early works, Mishra et al. (2018) fail to process temporal sequence with vanilla Transformers and find it even worse than random policy under certain tasks. Gated Transformer-XL (GTrXL) (Parisotto et al., 2020) is the first efficacious scheme to use Transformer as the memory. GTrXL provides a gated 'skip' path with Identity Map Reordering to stabilize the training procedure from the beginning. Such architecture can also incorporate language instructions to accelerate meta RL (Bing et al., 2022) and multi-task RL (Guhur et al., 2023). Furthermore, Loynd et al. (2020) propose a shortcut mechanism with memory vectors for long-term dependency, and Irie et al. (2021) combine the linear Transformer with Fast Weight Programmers for better performance. In addition, Melo (2022) proposes to use the self-attention mechanism to mimic memory reinstatement for memory-based meta RL. Esslinger et al. (2022) combine Transformer with Bellman loss to process observation history as the input of Q-network.

The attention mechanism in Transformers avoids the need for recurrent context input, making it superior to recurrent models in encoding long dependencies. While Transformer outperforms LSTM/RNN as the memory horizon grows and parameter scales, it suffers from poor data efficiency with RL signals. Follow-up works exploit some auxiliary (self-) supervised tasks to benefit learning (Banino et al., 2021) or use a pre-trained Transformer as a temporal encoder (Li et al., 2022; Fan et al., 2022).

### 4.2 Transformers for model learning

In addition to using Transformers as the encoder for sequence embedding, Transformer architecture also serves as the backbone of the world model in model-based algorithms. Distinct from the prediction conditioned on single-step observation and action, Transformer enables the world model to predict transition conditioned on historical information.

Practically, the success of Dreamer and subsequent algorithms (Hafner et al., 2020; 2021; 2023; Seo et al., 2022b) has demonstrated the benefits of world models conditioned on history in partially observable environments or in tasks that require a memory mechanism. A world model conditioned on history consists of an observation encoder to capture abstract information and a transition model to learn the transition in latent space, formally:

$$z_t \sim P_{\text{enc}}(z_t|o_t),$$
$$\hat{z}_{t+1}, \hat{r}_{t+1}, \hat{\gamma}_{t+1} \sim P_{\text{trans}}(\hat{z}_{t+1}, \hat{r}_{t+1}, \hat{\gamma}_{t+1}|z_{\leq t}, a_{\leq t}),$$

where $z_t$ represents the latent embedding of observation $o_t$, and $P_{\text{enc}}, P_{\text{trans}}$ denote observation encoder and transition model, respectively.

There are several attempts to build a world model conditioned on history with Transformer architecture instead of RNN in previous works. Concretely, Chen et al. (2022) replace the RNN-based Recurrent State-Space Model (RSSM) in Dreamer with a Transformer-based model (Transformer State-Space Model, TSSM). IRIS (Imagination with auto-Regression over an Inner Speech) (Micheli et al., 2022) and Transformer-based World Model (TWM) (Robine et al., 2023) learns a Transformer-based world model simply via auto-regressive learning on rollout experience without KL balancing like Dreamer and achieves considerable results on the Atari (Bellemare et al., 2013) 100k benchmark.

Besides, some works also try to combine Transformer-based world models with planning. Ozair et al. (2021) verify the efficacy of planning with a Transformer-based world model to tackle stochastic tasks requiring long tactical look-ahead. Sun et al. (2022) propose a goal-conditioned Transformer-based world model which is effective in visual-grounded planning for procedural tasks.

It is true that both RNN and Transformer are compatible with world models conditioned on historical information. However, Micheli et al. (2022); Chen et al. (2022) find that Transformer is a more data-efficient world model compared with Dreamer, and experimental results of TSSM demonstrate that Transformer architecture is lucrative in tasks that require long-term memory. In fact, although model-based methods are data-efficient, they suffer from the

| Method | Setting | Hindsight Info | Inference | Additional Structure/Usage |
|---|---|---|---|---|
| DT (Chen et al., 2021) | Offline | return-to-go | conditioning | basic Transformer structure |
| TT (Janner et al., 2021) | IL/GCRL/Offline | return-to-go | beam search | basic Transformer structure |
| BeT (Shafiullah et al., 2022) | BC | none | conditioning | basic Transformer structure |
| BooT (Wang et al., 2022) | Offline | return-to-go | beam search | data augmentation |
| GDT (Furuta et al., 2021) | HIM | arbitrary | conditioning | anti-causal aggregator |
| ESPER (Paster et al., 2022) | Offline (stochastic) | expected return | conditioning | adversarial clustering |
| DoC (Yang et al., 2022a) | Offline (stochastic) | learned representation | conditioning | additional latent value func. |
| QDT (Yamagata et al., 2022) | Offline | relabelled return-to-go | conditioning | additional Q func. |
| StARformer (Shang et al., 2022) | IL/Offline | return-to-go/reward | conditioning | Step Transformer & Sequence Transformer |
| TIT (Mao et al., 2022) | Online/Offline | return-to-go/none | conditioning | Inner Transformer & Outer Transformer |
| ConDT (Konan et al., 2022) | Offline | learned representation | conditioning | return-dependent transformation |
| SPLT (Villaflor et al., 2022) | Offline | none | min-max search | separate models for world and policy |
| DeFog (Hu et al., 2023) | Offline | return-to-go | conditioning | drop-span embedding |
| ODT (Zheng et al., 2022) | Online finetune | return-to-go | conditioning | trajectory-based entropy |
| MADT (Meng et al., 2021) | Online finetune (multi-agent) | none | conditioning | separate models for actor and critic |

Table 1: A summary of Transformers for sequential decision-making.

compounding prediction error increasing with model rollout length, which greatly affects the performance and limits model rollout length (Janner et al., 2019). The Transformer-based world model can help alleviate the prediction error on longer sequences (Chen et al., 2022; Janner et al., 2021).

## 4.3 Transformers for sequential decision-making

In addition to being an expressive architecture to be plugged into components of traditional RL algorithms, Transformer itself can serve as a model that conducts sequential decision-making directly. This is because RL can be viewed as a conditional sequence modeling problem — generating a sequence of actions that can yield high returns.

### 4.3.1 Transformers as a milestone for offline RL

One challenge for Transformers to be widely used in RL is that the non-stationarity during the training process may hinder its optimization. However, the recent prosperity in offline RL motivates a growing number of works focusing on training a Transformer model on offline data that can achieve state-of-the-art performance. Decision Transformer (DT) (Chen et al., 2021) first applies this idea by modeling RL as an autoregressive generation problem to produce the desired trajectory:

$$\tau = \left( \hat{R}_1, s_1, a_1, \hat{R}_2, s_2, a_2, \ldots, \hat{R}_T, s_T, a_T \right),$$

where $\hat{R}_t = \sum_{t'=t}^{T} r(s_{t'}, a_{t'})$ is the return-to-go. By conditioning on proper target return values at the first timestep, DT can generate desired actions without explicit TD learning or dynamic programming. Concurrently, Trajectory Transformer (TT) (Janner et al., 2021) adopts a similar autoregressive sequence modeling:

$$\log P_\theta(\tau_t|\tau_{<t}) = \sum_{i=1}^{N} \log P_\theta(s_t^i|s_t^{<i}, \tau_{<t}) + \sum_{j=1}^{M} \log P_\theta(a_t^j|a_t^{<j}, s_t, \tau_{<t}) + \log P_\theta(r_t|a_t, s_t, \tau_{<t}) + \log P_\theta(\hat{R}_t|r_t, a_t, s_t, \tau_{<t}),$$

where $M$ and $N$ denote the dimension of state and action, respectively. In contrast to selecting target return values, TT proposes to use beam search for planning during execution. The empirical results demonstrate that TT performs well on long-horizon prediction. Moreover, TT shows that with mild adjustments on vanilla beam search, TT can perform imitation learning, goal-conditioned RL, and offline RL under the same framework. Regarding the behavior cloning setting, Behavior Transformer (BeT) (Shafiullah et al., 2022) proposes a similar Transformer structure as TT to learn from multi-modal datasets.

In light of Transformer's superior accuracy on sequence prediction, Bootstrapped Transformer (BooT) (Wang et al., 2022) proposes to bootstrap Transformer to generate data while optimizing it for sequential decision-making. Formally, given a trajectory $\tau$ from the original dataset, BooT resamples the last $T' < T$ timesteps, and concatenates the generated $T'$ steps with the original $T - T'$ steps as a new trajectory $\tau'$:

$$\tau' = \left( \tau_{\leq T-T'}, \tilde{\tau}_{>T-T'} \right), \quad \tilde{\tau}_{>T-T'} \sim P_\theta(\tau_{>T-T'}|\tau_{\leq T-T'}).$$

Bootstrapping Transformer for data augmentation can expand the amount and coverage of offline datasets, and hence improve the performance. More specifically, this work compares two different schemes to predict the next token $\tilde{y}_n$:

$$\tilde{y}_n \sim P_\theta(y_n|\tilde{y}_{<n}, \tau_{\leq T-T'}); \quad \text{(autoregressive generation, sampling conditioned on previously } \textit{generated} \text{ tokens)}$$
$$\tilde{y}_n \sim P_\theta(y_n|y_{<n}, \tau_{\leq T-T'}), \quad \text{(teacher-forcing generation, sampling conditioned on previously } \textit{original} \text{ tokens)}$$

and the results show that using two schemes together can generate data consistent with the underlying MDP without additional explicit conservative constraints.

### 4.3.2 Different choices of conditioning

While conditioning on return-to-go is a practical choice to incorporate future trajectory information, one natural question is whether other kinds of hindsight information can benefit sequential decision-making. To this end, Furuta et al. (2021) propose Hindsight Information Matching (HIM), a unified framework that can formulate variants of hindsight RL problems. More specifically, HIM converts hindsight RL into matching any predefined statistics of future trajectory information w.r.t. the distribution induced by the learned conditional policy:

$$\min_\pi \mathbb{E}_{z \sim p(z), \tau \sim \rho_z^\pi(\tau)}[D(I^\Phi(\tau), z)],$$

where $z$ is the parameter that policy $\pi(a|s, z)$ is conditioned on, $D$ is a divergence measure such as KL divergence or f-divergences, and the information statistics $I^\Phi(\tau)$ can be any function of the trajectory via the feature function $\Phi$. Furthermore, this work proposes Generalized DT (GDT) for arbitrary choices of statistics $I^\Phi(\tau)$[3] and demonstrates its applications in two HIM problems: offline multi-task state-marginal matching and imitation learning.

Specifically, one drawback of conditioning on return-to-go is that it will lead to sub-optimal actions in stochastic environments, as the training data may contain sub-optimal actions that luckily result in high rewards due to the stochasticity of transitions. Paster et al. (2022) identify this limitation in general RvS methods. They further formulate RvS as an HIM problem and discover that RvS policies can achieve goals in consistency if the information statistics are independent of transitions' stochasticity. Based on this implication, they propose environment-stochasticity-independent representations (ESPER), an algorithm that first clusters trajectories and estimates average returns for each cluster, and then trains a policy conditioned on the expected returns. Alternatively, Dichotomy of Control (DoC) (Yang et al., 2022a) proposes to learn a representation that is agnostic to stochastic transitions and rewards in the environment via minimizing mutual information. During inference, DoC selects the representation with the highest value and feeds it into the conditioned policy.

Yang et al. (2022b) proposed a method by annotating task-specific procedure observation sequences in the training set and using them to generate decision actions. This approach enables the agent to learn decision-making based on prior planning, which is beneficial for tasks that require multi-step foresight.

In addition to exploring different hindsight information, another approach to enhance return-to-go conditioning is to augment the dataset. Q-learning DT (QDT) (Yamagata et al., 2022) proposes to use a conservative value function to relabel return-to-go in the dataset, hence combining DT with dynamic programming and improving its stitching capability.

### 4.3.3 Improving the structure of Transformers

Apart from studying different conditioned information, there are some works to improve the structure of DT or TT. These works fit into two categories: **a) Introduce additional modules or loss functions to improve the representation.** Shang et al. (2022) argue that the DT structure is inefficient for learning Markovian-like dependencies, as it takes all tokens as the input. To alleviate this issue, they propose learning an additional Step Transformer for local state-action-reward representation and using this representation for sequence modeling. Similarly, Mao et al. (2022) use an inner Transformer to process the single observation and an outer Transformer to process the history, and cascade them as a backbone for online and offline RL. Konan et al. (2022) believe that different sub-tasks correspond to different levels of return-to-go, which require different representation. Therefore, they propose Contrastive Decision Transformer (ConDT) structure, where a return-dependent transformation is applied to state and action embedding

---

[3]As a special case, $I^\Phi(\tau) = \hat{R}_t$ in DT.

before putting them into a causal Transformer. The return-dependent transformation intuitively captures features specific to the current sub-task, and it is learned with an auxiliary contrastive loss to strengthen the correlation between transformation and return. **b) Design the architecture to improve the robustness.** Villaflor et al. (2022) identify that TT structure may produce overly optimistic behavior, which is dangerous in safety-critical scenarios. This is because TT implements model prediction and policy network in the same model. Therefore, they propose SeParated Latent Trajectory Transformer (SPLT Transformer), which consists of two independent Transformer-based CVAE structures of the world model and policy model, with the trajectory as the condition. Formally, given the trajectory $\tau_t^K = (s_t, a_t, s_{t+1}, a_{t+1}, \ldots, s_{t+K}, a_{t+K})$ and its sub-sequence $\tau'^k_t = (s_t, a_t, s_{t+1}, a_{t+1}, \ldots, s_{t+k})$, SPLT Transformer optimizes two variational lower bounds for the policy model and the world model:

$$\mathbb{E}_{z_t^\pi \sim q_{\phi_\pi}} \left[ \sum_{k=1}^{K} \log p_{\theta_\pi}(a_{t+k}|\tau'^k_t; z_t^\pi) \right] - D_{\text{KL}}(q_{\phi_\pi}(z_t^\pi|\tau_t^K), p(z_t^\pi)),$$

$$\mathbb{E}_{z_t^w \sim q_{\phi_w}} \left[ \sum_{k=1}^{K} \log p_{\theta_w}(s_{t+k+1}, r_{t+k}, \hat{R}_{t+k+1}|\tau_t^k; z_t^w) \right] - D_{\text{KL}}(q_{\phi_w}(z_t^w|\tau_t^K), p(z_t^w)),$$

where $q_{\phi_\pi}$ and $p_{\theta_\pi}$ are the encoder and decoder of the policy model, and $q_{\phi_w}$ and $p_{\theta_w}$ are the encoder and decoder of the world model, respectively. Similarly to the min-max search procedure, SPLT Transformer searches the latent variable space to minimize return-to-go in the world model and to maximize return-to-go in the policy model during planning. Hu et al. (2023) consider the possible frame dropping in the actual application scenario where states and rewards at some timesteps are unavailable due to frame dropping and the information at previous timesteps is left to be reused. They propose Decision Transformer under Random Frame Dropping (DeFog), a DT variant which extends the timestep embedding by introducing an additional drop-span embedding to predict the number of consecutive frame drops. In addition to frame dropping, DT may also suffer from forgetting, as it merely relies on parameters to "memorize" the large-scale data in an implicit way. Therefore, Kang et al. (2023) introduce an internal working memory module into DT to extract knowledge from past experience explicitly. They also incorporate low-rank adaptation (LoRA) (Hu et al., 2022) parameters to adapt to unseen tasks.

### 4.3.4 Extending DT beyond offline RL

Although most of the works around Transformers for sequential decision-making focus on the offline setting, there are several attempts to adapt this paradigm to online and multi-agent settings. Online Decision Transformer (ODT) (Zheng et al., 2022) replaces the deterministic policy in DT with a stochastic counterpart and defines a trajectory-level policy entropy to help exploration during online fine-tuning. Pretrained Decision Transformer (PDT) (Xie et al., 2022) adopts the idea of Bayes' rule in online fine-tuning to control the conditional policy's behavior in DT. Besides, such a two-stage paradigm (offline pre-training with online fine-tuning) is also applied to Multi-Agent Decision Transformer (MADT) (Meng et al., 2021), where a decentralized DT is pre-trained with offline data from the perspective of individual agents and is used as the policy network in online fine-tuning with MAPPO (Yu et al., 2021a). Interestingly, to the best of our knowledge, we do not find related works about learning DT in the pure online setting without any pre-training. We suspect that this is because pure online learning will exacerbate the non-stationary nature of training data, which will severely harm the performance of the current DT policy. Instead, offline pre-training can help to warm up a well-behaved policy that is stable for further online training.

### 4.4 Transformers for generalist agents

In view of the fact that Decision Transformer has already flexed its muscles in various tasks with offline data, several works turn to consider whether Transformers can enable a generalist agent to solve multiple tasks or problems, as in the CV and NLP fields.

### 4.4.1 Generalize to multiple tasks

**Some works draw on the ideas of pre-training on large-scale datasets in CV and NLP, and try to abstract a general policy from large-scale multi-task datasets.** Multi-Game Decision Transformer (MGDT) (Lee et al., 2022), a variant of DT, learns DT on a diverse dataset consisting of both expert and non-expert data and achieves close-to-human performance on multiple Atari games with a single set of parameters. In order to obtain expert-level performance with

a dataset containing non-expert experiences, MGDT includes an expert action inference mechanism, which calculates an expert-level return-to-go posterior distribution from the prior distribution of return-to-go and a preset expert-level return-to-go likelihood proportional according to the Bayesian formula. Likewise, Switch Trajectory Transformer (SwitchTT) (Lin et al., 2022), a multi-task extension to TT, exploits a sparsely activated model that replaces the FFN layer with a mixture-of-expert layer for efficient multi-task offline learning. More specifically, such "switch layer" consists of $n$ expert networks $E_1, \ldots, E_n$ and a router to select a specific expert for each token:

$$y = p_i(x)E_i(x), \quad i = \arg\max_i p_i(x) = \arg\max_i \frac{e^{h(x)_i}}{\sum_j^n e^{h(x)_j}},$$

where $h(x)$ denotes the logits produced by the router. Besides, a distributional trajectory value estimator is adopted to model the uncertainty of value estimates. With these two enhanced features, SwitchTT achieves improvement over TT across multiple tasks in terms of both performance and training speed. MGDT and SwitchTT exploit experiences collected from multiple tasks and various performance-level policies to learn a general policy. To solve multi-objective RL problem, Zhu et al. (2023) build a multi-objective offline RL dataset and extend DT to preference and return conditioned learning. As for meta RL across diverse tasks, Team et al. (2023) demonstrate that Transformer model-based RL can improve adaptation and scalability with curriculum and distillation.

However, constructing a large-scale multi-task dataset is non-trivial. Unlike large-scale datasets in CV or NLP, which are usually constructed with massive public data from the Internet and simple manual labeling, action information is always absent from public sequential decision-making data and is not facile to label. Thus, Baker et al. (2022) propose a semi-supervised scheme to utilize large-scale online data without action information by learning a Transformer-based Inverse Dynamic Model (IDM) on a small portion of action-labeled data, which predicts the action information with past and future observations and is consequently capable of labeling massive unlabeled online video data. IDM is learned on a small-scale dataset containing manually labeled actions and is accurate enough to provide action labels of videos for effective behavior cloning and fine-tuning. Further, Venuto et al. (2022) conduct experiments where they train IDM with action-labeled data from other tasks, which reduces the need of action-labeled data specifically for the target task.

**The efficacy of prompting (Brown et al., 2020) for adaptation to new tasks has been proven in many prior works in NLP. Following this idea, several works aim at leveraging prompting techniques for DT-based methods to enable fast adaptation.** Prompt-based Decision Transformer (Prompt-DT) (Xu et al., 2022) samples a sequence of transitions from the few-shot demonstration dataset as prompt, and shows that it can achieve few-shot policy generalization on offline meta RL tasks. Reed et al. (2022) further exploit prompt-based architecture to learn a generalist agent (Gato) via auto-regressive sequence modeling on a super large-scale dataset covering natural language, image, temporal decision-making, and multi-modal data. Gato is capable of a range of tasks from various domains, including text generation and decision-making. Specifically, Gato unifies multi-modal sequences in a shared tokenization space and adapts prompt-based inference in deployment to generate task-specific sequences. Similarly to Gato, RT-1 (Brohan et al., 2022) and PaLM-E (Driess et al., 2023) leverage large-scale multimodal datasets to train Transformers that can achieve high performance on downstream tasks. VIMA (Jiang et al., 2022) combines textual and visual tokens as multimodal prompts to build a scalable model that can generalize well among robot manipulation tasks.

Despite being effective, Laskin et al. (2022) point out that one limitation of the prompt-based framework is that the prompt is demonstrations from a well-behaved policy, as contexts in both works are not sufficient to capture policy improvement. Inspired by the in-context learning capabilities (Alayrac et al., 2022) of Transformers, they propose Algorithm Distillation (AD) (Laskin et al., 2022), which instead trains a Transformer on across-episode sequences of the learning progress of single-task RL algorithms. Therefore, even in new tasks, the Transformer can learn to gradually improve its policy during the auto-regressive generation.

### 4.4.2 Generalize to multiple domains

Beyond generalizing to multiple tasks, Transformer is also a powerful "universal" model to unify a range of domains related to sequential decision-making. Motivated by advances in masked language modeling (Devlin et al., 2018) technique in NLP, Carroll et al. (2022) propose Uni[MASK], which unifies various commonly-studied domains, including behavioral cloning, offline RL, GCRL, past/future inference, and dynamics prediction, as one mask inference problem. Uni[MASK] compares different masking schemes, including task-specific masking, random masking, and

finetune variants. It is shown that one single Transformer trained with random masking can solve arbitrary inference tasks. More surprisingly, compared to the task-specific counterpart, random masking can still improve performance in the single-task setting.

In addition to unifying sequential inference problems in the RL domain, Reid et al. (2022) find it beneficial to fine-tune DT with Transformer pre-trained on language datasets or multi-modal datasets containing language modality. Concretely, Reid et al. (2022) find out that pre-training Transformer with language data while encouraging similarity between language and RL-based representations can help improve the performance and convergence speed of DT. This finding implies that even knowledge from non-RL fields can benefit RL training via Transformers. Li et al. (2022) further show that a policy initialized with pre-trained language models can be finetuned to accommodate different tasks and goals in interactive decision-making. Additionally, several works (Huang et al., 2022a;b; Raman et al., 2022; Yao et al., 2022; Ahn et al., 2022; Wang et al., 2023c; Du et al., 2023; Wang et al., 2023a) discover that pre-trained large-scale language models are capable of generating reasonable high-level plans to accomplish complex tasks without further finetuning. However, even with well-behaved low-level policies, directly applying large language models for execution is inefficient or even infeasible in particular scenarios or environments. Therefore, these works propose to generate valid action sequences with affordance guidance (Ahn et al., 2022), autoregressive correction (Huang et al., 2022a), corrective re-prompting (Raman et al., 2022), interleaved reasoning and action traces generation (Yao et al., 2022), and description feedback (Huang et al., 2022b; Wang et al., 2023c; Du et al., 2023; Wang et al., 2023a). Even without RL modules, Wu et al. (2023) find that GPT-4 (OpenAI, 2023) , when prompted with related RL paper and chain-of-thought, can produce a competitive performance in RL benchmark tasks.

## 5 Summary and Future Perspectives

This paper briefly reviews advances in Transformers for RL. We provide a taxonomy of these advances: *a)* Transformers can serve as a powerful module of RL, e.g., acting as a representation module or a world model; *b)* Transformers can serve as a sequential decision-maker; and *c)* Transformers can benefit generalization across tasks and domains. While we cover representative works on this topic, the usage of Transformers in RL is not limited to our discussions. Given the prosperity of Transformers in the broader AI community, we believe that combining Transformers and RL is a promising trend. To conclude this survey, in the following, we discuss future perspectives and open problems for this direction.

**Bridging Online and Offline Learning via Transformers.** Stepping into offline RL is a milestone in Transformer-based RL. Practically, utilizing Transformers to capture dependencies in decision sequence and to abstract policy is mainly inseparable from the support of considerable offline demonstrations. However, it is unfeasible for some decision-making tasks to get rid of the online framework in real applications. On the one hand, it is not that easy to obtain expert data in certain tasks while a substantial amount of less related data holds potential valuable information. On the other hand, some environments are open-ended (e.g., Minecraft), which means the policy has to continually adjust to deal with unseen tasks during online interaction. Therefore, we believe that bridging online learning and offline learning is necessary. However, most research progress following Decision Transformer focuses on the offline learning framework. We thereby expect some special paradigm designs to address this issue via transferring the generalization capabilities demonstrated by Transformer structures in large vision or language models to decision tasks and to potentially significantly enhancing the utilization of less related offline data. In addition, how to train a well-performed online Decision Transformer from scratch is an interesting open problem.

**Combining Reinforcement Learning and (Self-) Supervised Learning.** Retracing the development of Transformer-based RL, the training methods involve both RL and (self-) supervised learning. When trained under the conventional RL framework, the Transformer architecture is usually unstable for optimization (Parisotto et al., 2020). RL algorithms inherently require imposing various constraints to avoid issues like the deadly triad (Van Hasselt et al., 2018), which could potentially exacerbate the training difficulty of the Transformer architecture. Meanwhile, the (self-) supervised learning paradigm, while improving the optimization of the Transformer structure, significantly constrains the performance of the policy based on the quality of the experience/demonstration data. Therefore, a better policy may be learned when we combine RL and (self-) supervised learning in Transformer learning. Some works (Zheng et al., 2022; Meng et al., 2021) have tried out the scheme of supervised pre-training and RL-involved fine-tuning. However, exploration can be limited with a relatively fixed policy (Nair et al., 2020), which is one of the

bottlenecks to be resolved. Also, along this line, the tasks used for performance evaluation are relatively simple. It is worthwhile to further explore whether Transformers can scale such (self-) supervised learning up to larger datasets, more complex environments, and real-world applications. Further, we expect future work to provide more theoretical and empirical insights to characterize under which conditions this (self-) supervised learning is expected to perform well (Brandfonbrener et al., 2022; Siebenborn et al., 2022; Takagi, 2022).

**Transformer Structure Tailored for Decision-making Problems.** The Transformer structures in the current DT-based methods are mainly vanilla Transformer, which is originally designed for the text sequence and may not fit the nature of decision-making problems. For example, is it appropriate to adopt the vanilla self-attention mechanism for trajectory sequences? Whether different elements in the decision sequence or different parts of the same elements need to be distinguished in position embedding? In addition, as there are many variants of representing trajectory as a sequence in different DT-based algorithms, how to choose from them still lacks systematic research. For instance, how to select robust hindsight information when deploying such algorithms in the industry? Furthermore, the vanilla Transformer is a structure with huge computational cost, which makes it expensive at both training and inference stages, and high memory occupation, which constrains the length of the dependencies it captures. We notice that some works (Zhou et al., 2021; Fedus et al., 2022; Zhu et al., 2021) have improved the structure from these aspects and have been validated in various domains. It is also worth exploring whether similar structures can be used in the decision-making problem.

**Towards More Generalist Agents with Transformers.** Our review has shown the potential of Transformers as a general policy (Section 4.4). In fact, the design of Transformers allows multiple modalities (e.g., image, video, text, and speech) to be processed using similar processing blocks and demonstrates excellent scalability to large-capacity networks and huge datasets. Recent works have made substantial progress in training agents capable of performing multiple and cross-domain tasks. However, given that these agents are trained on huge amounts of data, it is still uncertain whether they merely memorize the dataset and if they can perform efficient generalization. Therefore, how to learn an agent that can generalize to unseen tasks without strong assumptions is well worth studying (Boustati et al., 2021). Moreover, we are curious about whether Transformer is strong enough to learn a general world model (Schubert et al., 2023) for different tasks and scenarios.

**Connections to Other Research Trends.** While we have discussed how RL can benefit from the usage of Transformers, the reverse direction, i.e., using RL to benefit Transformer training is an intriguing open problem yet less explored. We see that, some works model language/dialogue generation tasks with offline RL setting and learn generation policy with relabeling (Snell et al., 2022b) or value functions (Verma et al., 2022; Snell et al., 2022a; Jang et al., 2022). Recently, Reinforcement Learning from Human Feedback (RLHF) (Ouyang et al., 2022) learns a reward model and uses RL algorithms to finetune Transformer for aligning language models with human intent (Nakano et al., 2021; OpenAI, 2023). In the future, we believe RL can be a useful tool to further polish Transformer's performance in other domains. In parallel with Transformers, diffusion model (Song et al., 2020; Ho et al., 2020) is also becoming a mainstream tool for generative tasks and demonstrates its applications in RL (Janner et al., 2022). While no significant performance gap is observed between Transformer-based RL and Diffusion-based RL (Janner et al., 2019; Ajay et al., 2022), we speculate that each model may have advantages in the domain they are most commonly used, i.e., Transformer in language RL tasks and diffusion model in vision RL tasks. Moreover, diffusion models may perform better in stochastic environments as they can model complex distributions. However, the relatively slow inference may hinder diffusion models' usage in real-time decision-making problems. We hope future work can conduct rigorous experiments and provide more empirical evidence to verify our conjectures.

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
