# OpenReview forum: "A Survey on Transformers in Reinforcement Learning"
_TMLR — Accepted by TMLR_

### Review · Reviewer_XCTM · 2023-07-16

**Summary Of Contributions:**

Main contributions:
- This survey provides a taxonomy of different applications of the transformer architectures in RL, including transformer as state encoder, as world model, or as decision maker (in the decision-transformer style).
- The authors also reviewed works that apply transformer models to train generalist agents, either from scratch or by fine-tuning or prompting models.
- The authors discussed future perspectives and open problems.

**Audience:**

Yes

**Broader Impact Concerns:**

No concerns

**Claims And Evidence:**

Yes

**Requested Changes:**

I only have a few non-critical requests:

Inaccurate description of cited papers:
- In section 3.2, "Emmons et al. (2021) identify that carefully choosing ... are crucial for the performance of DRL agents." That paper only discusses offline RL.
- In section 5, under "Transformer Structure Tailored for Decision-making Problems": Zhou et al., 2021 is not an NLP paper. If the authors want to mention efficient transformer variants, Tay et al., 2022 (already cited) should have pointers to representative works.

Questionable claims
- In the introduction, "the training objective of RL agents is typically a function of the current policy, which induces nonstationarity
during learning a Transformer". The non-stationary is not caused by using the policy in the training objective. It's caused by the training data being collected using an ever-changing policy.
- Section 4.2, "The Transformer-based world model can help alleviate the prediction error on longer sequences." Please add citations to support this claim.

A few missing citations
- When discussing architectures in DRL, please consider cite IMPALA [1]
- Deep Transformer Q-Networks [2] uses transformer with observation history as input, and the bellman loss is trained for multiple time-steps together.
- [3] discusses applying ViT to RL

Typos:
- Section 4.4.2: "directly applying large language models for execution is inefficient or even feasible". feasible -> infeasible or unfeasible.

Additional suggestion:
-  In the introduction and section 3.2, the authors discuss a few challenges of adopting transformer to RL problems. I think another possible reason is that transformer models typically need much more training data to be effective compared to models with strong inductive biases (CNN, RNN). In traditional DRL, sample efficiency is a key benchmark, which limits how much data the agent can collect to fit the model. It might be helpful to discuss this aspect as well.

[1] IMPALA: Scalable Distributed Deep-RL with Importance Weighted Actor-Learner Architectures. Espeholt et al., 2018

[2] Deep transformer q-networks for partially observable reinforcement learning. Esslinger et al., 2022

[3] Stabilizing Deep Q-Learning with ConvNets and Vision Transformers under Data Augmentation. Hansen et al., 2021



**Strengths And Weaknesses:**

Strength
- This is a timely survey for an emerging and promising research area. The works mentioned in the paper are comprehensive and representative, and cover all the main areas (as far as I know) in this filed.
- The authors provide a brief historical perspective on the development of model architectures in RL, and discussed challenges in applying transformer in the RL domain, e.g. the problem of non-stationary training data. This sets up an adequate context for the main discussion.

Weakness (For each bullet, please see "requested changes" for details.)
- There are a few inaccurate description of cited papers.
- A few claims are not well supported or need rephrasing.
- A few missing citations.

---

### Review · Reviewer_e3ki · 2023-07-19

**Summary Of Contributions:**

This paper is a survey on the use of Transformers in Deep Reinforcement Learning. After discussing challenges of the use of such architectures (notably related w.r.t. to their need of massive training data and their latency), authors present works for different use of Transformers, namely for representation learning in RL, model learning and as the actor of the learned policy. The latter is the most developed, mainly in the scope of offline learning. Finally authors discuss advantages of Transformers for learning generalist agents and give some insights about their view on important future works in that scope.

**Audience:**

Yes

**Broader Impact Concerns:**

.

**Claims And Evidence:**

Yes

**Requested Changes:**

- I would really appreciate if authors could give more details at least about the following approaches : GDT, SPLT, SwitchTT, BooT. Giving some formalized components of their principles.
- A trending emerging architecture is decision diffusers for RL tasks. It would have been useful if authors could discuss pros and cons of transformers w.r.t. these new competitors.
- develop section 4.3.4
- I have difficulties to understand the core differences between the two first paragraphs given as perspectives in 5. For me it is very close...
- Authors discuss in two different places about the "deadly triad problem" that hinders the use of Transformers for online learning  of RL policies. In the first para of perspectives (section 5), it is argued that self-supervised learning of sequences is thus better. I would like to point out that the policy gradient paradigm is fully equivalent to such a log-likelihood maximization when all rewards are set to 1 in every trajectory. For me, appart the onpolicy exploration process of policy gradients,  there is no real difference between both paradigms. So this does not look very "new". Could authors elaborate on this ? By the way, policy gradient is fully ignored in the survey...
- The last perspective paragraph (RL for transformers) should be removed : while it suggests that RL could be used to learn better transformer architectures, the discussion is about the use of Transformers  for sequences modelling, which has no difference with imitation learning (and RLHF is simply RL finetuning as it is done for any other decision-making problem, I cannot understand why authors treat that tasks differently here).

**Strengths And Weaknesses:**

Strengths:
  - many important recent works are listed
  - the paper makes the reader questioning about challenges and potential of these architectures
  - some interesting discussions

Weakness :
   - No experiments to support discussions
   - Not enough details in many places. It would have been very useful to choose one important approach in each section, to be detailed, or at least a bit formalized (e.g., GDT, SPLT, SwitchTT, BooT, etc). It would greatly help the reader to organize the given information, which is a bit dense in the current form, and focus on main aspects of the discussion (I was a bit lost in the textual flow, some illustrations and equations usually help to clarify).
- Sometimes a lack of structure. Giving bold titles to paragraphs could help gathering the main idea of each. Section 4.4.1 is an example of section that would greatly benefit from a  more highlighted structure.
- Difficult to understand why challenges given in the paper are specific to transformers. Need of massive training data and latency are well-known challenges related to any deep architectures (including eg RNN)
 - The survey devotes a very large part to offline learning, specifically following decision transformers (and GDT) principles. While it is true that it is mainly used in that setting, it would have been very useful to discuss further the reasons that limit their  applicability in online RL (for instance by giving return-to-go as input as DT does for offline data). Section 4.3.4 is really too short on this, please develop.

---

### Review · Reviewer_KsT8 · 2023-07-25

**Summary Of Contributions:**

Transformers have emerged as a general and powerful model for learning from high-dimensional, rich, and sequential data that has seen success across variety of domains of ML, including NLP and Computer Vision. This paper surveys the use of Transformers in Reinforcement Learning, categorizing uses of transformers into the following:

1. as one component of a more traditional RL agent system, e.g. as an observation encoder, especially when processing rich visual or hierarchical/abstract data; as a memory module that processes past states (as an alternative to other memory mechanisms such as RNNs); or as a world model;
2. as an "end-to-end" sequential decision maker, by treating RL as a sequence modeling problem, in the line of decision transformer and other approaches to hindsight RL;
3. as very large, generalist models that can solve many tasks and domains, especially when combined with pretrained models such as large language models

Within these three taxonomies, the authors survey a fairly comprehensive set of recent work, including core ideas (e.g. RL as sequence modeling, generalist models), as well as additional work attempting to build and generalize such ideas. They conclude with several promising directions for future research.


**Audience:**

Yes

**Claims And Evidence:**

Yes

**Requested Changes:**

As I described above, I would be happy to be overriden if my co-reviewers believe the paper is fine as is, but I think to be an excellent review paper, the paper could (1) slightly change the wording to be more clear about the different roles that Transformers play across different recent advances in RL, and (2) more clearly explain the advantages and shortcomings of the Transformer architecture specifically (rather than listing challenges specific to RL as a whole), and how recent work (e.g. section 4.3.3) is trying to get around such limitations.

**Strengths And Weaknesses:**

# Strengths

- This area is clearly of interest to many in the TMLR audience, as transformers are increasingly the go-to modeling architecture of choice across a variety of ML domains, including RL. This paper should serve as a reasonably recent and comprehensive reference for practitioners
- I largely agree with the authors' taxonomy of uses of various Transformer architectures, though I disagree with the framing (see Weaknesses)
- Good survey of future directions and outstanding challenges in the field.

# Weaknesses

## Framing

I feel conflicted about the paper—on the one hand, Transformers have upended RL (and every other ML subfield) and so it's obviously important for people to understand how they're used in RL, and towards that end this paper does a good job at surveying the recent work and latest advancements (it may have missed a few papers, since it's hard to catch every paper that gets released in this field, but it feels reasonably comprehensive). In providing a reasonably comprehensive overview of the recent literature, this paper does a good job.

On the other hand, I find the framing of the paper a bit reductive, as it simplifies and combines several distinct research advances and thrusts under the generic umbrella of "use Transformers for RL". Specifically, the authors identify three "pillars" of recent advances in RL:

1. Using Transformers to model high-dimensional, abstract sequential and/or graph-structured data as components in more "traditional" RL systems.
2. The view of reinforcement learning as sequential decision making + the hindsight information matching framework. Note that this advance is *not* inherent to Transformers, though obviously the first work pushing this view indeed used a Transformer as its modeling architecture of choice.
3. The gradual trend towards large-scale, generalist, pretrained ("foundation") models that learn to transfer and share parameters across a variety of tasks for improved performance, and the use of already successful pretrained models (such as large language models) as backbones for RL systems. Note that this advance is *also* not specific to Transformer models, though obviously the overwhelming majority of currently successful foundation models are Transformer-based.

Note that of these three thrusts, all 3 involve transformers to some degree, but arguably only one of these (the first pillar) is actually specific to the Transformer architecture. The other I do believe should be included in a Transformers for RL review paper, **however**  I find the paper takes a bit of a reductive framing by describing everything as Transformer-first. For example, section 4.3 is described as "using Transformers for sequential decision making", and is framed primarily as "let's use Transformers to solve a SDM problem." OTOH I believe the framing should be something like: "A recent advance in RL has been viewing RL as sequence modeling...Transformers are a particularly useful architecture for this approach because they are particularly good sequence models." Similarly, I feel like Section 4.4 is less about "Transformers for generalist agents" and more about "Transformers as (one choice) of a model architecture, towards the general trend of large-scale pretrained models and transfer learning for RL."

This framing can be seen in the authors' term "TransformRL". I disagree with the authors' decision to define a new name, TransformRL, to refer to the use of Transformers in RL, given the discussion on reductiveness above, and especially given how many overloaded and confusing terms the fields of RL/ML/DL have already. In my opinion the term conflates the field (RL) with current tools that we find useful to use in the field (Transformers) and the recent central advances dominating the RL conversation, of which Transformers play varying degrees of importance (RL as sequence modeling, large-scale pretraining). It's possible Transformer hype will die down, yet these other ideas will persist, and I don't think that RL should currently be defined by the use of Transformers. Imagine, for example, if we had analogous terms like RNNNLP or CNNCV—I view them all as rather unnecessary.

I think to improve the framing under my view, the paper could use a more nuanced discussion at the beginning, by saying that transformers are now a workhorse in RL because they are such expressive and capable models, but they are used for very (different) reasons stemming from orthogonal advances in RL, including (1) RL in high-dimensional spaces, (2) RL as a sequence modeling problem, and (3) the trend towards large-scale, pretrained foundation models.

## Could better summarize advantages and disadvantages *specific to Transformers* in RL

I find the paper could be more detailed and nuanced when describing the reasons why one might specifically use Transformers over other models in RL, and what features of problems make Transformers a particularly good choice. Most of the sections of the paper basically show that Transformers have seen success when slotting them into basically every possible part of the RL problem: when processing observations (section 4.1.1), memory (section 4.1.2), world models (section 4.1.3). The authors catalog dozens upon dozens of apparent success stories where Transformers successfully replace existing deep architectures for RL.

What should the takeaway be for the readers reading this paper? Is it that one should now use  Transformers basically unilaterally, in every possible component of an RL model? The paper could emphasize *particular problems* of an RL task that make Transformers a particularly suitable solution.

For example, in Section 4.1, it seems like Transformers have seen success in processing "complex information", including multi-agent representations (zambaldi et al 2018), morphological domain knowledge (kurin 2020), "sensory sequences" (tang and ha), multimodal information (team 2021), and a variety of vision settings. What is the overall recommendation or understanding from the authors, after surveying this work? Is it that one should no longer use any other more domain specific architectures such as CNNs and RNNs? Does it seem like Transformers be the go-to architecture for representation learning in RL? Are there any commonalities among domains where Transformers consistently outperform or underperform, or is more empirical work needed? I imagine that this work shows that Transformers have the most success in RL when applied the same kinds of representation learning problems as the supervised learning problems that Transformers are so good at. Assuming this is the case, the authors could more clearly illustrate this by unifying the case studies in this section, and tying it back to the general advantages of transformers as mentioned e.g. in 3rd paragraph of the intro (long dependencies, scalability).

Similarly, why should one use a Transformer instead of a more traditional memory mechanism such as an RNN/LSTM in RL agents (section 4.1.2)? Should it only be for environments with extremely long-distance dependencies? How should one think about the tradeoffs of (1) increased sequence modeling capacity and (2) larger model capacity? There is some description of the relevant tradeoffs in the final paragraph, but the writing could try to better aggregate and summarize the trends in each subarea here.

---

The paper also lacks nuance in describing the current outstanding challenges in using Transformers in RL. For example, end of page 1/beginning of page 2, and Section 3.2 describes "unique challenges" for Transformers in RL:
1. Sensitivity to architecture selection
2. Sensitivity to design choices (e.g. hyperparameters)
3. Large memory footprints and high latency

In my view, only (3) is a challenge that is arguably somewhat unique to Transformers, especially the large-scale transformers that have seen success across vision, NLP, and RL tasks. (1) and (2) seem like general issues across the entire field of RL.

## Minor

- I don't think "foundation model" is used correctly e.g. bottom of page 1, first sentence of section 4—I don't think a particular architecture can be categorized as a "foundation model" since you can still have small domain-specific transformers. Whether something is a "foundation model" is primarily determined by the model capabilities and usage intent rather than the particular architecture
- I don't think Transformers are strictly sequence models, since sequence implies linear order and one can apply transformers to various richly ordered or unordered representation learning problems. However section 4.1 seems to cast Transformers as such. For example it's not clear to me that all of the uses of Transformers in section 4.1.1 are **sequence encoding** problems—e.g. the multi-entity StarCraft modeling problem is not necessarily a sequence modeling problem but rather a set representation learning problem.
- End of section 4.2 "The Transformer-based world model can help alleviate the prediction error on longer sequences." Can authors elaborate? Is this an empirical finding from the Janner paper or just a general statement about improved sequence modeling capacity from transformers?
- I think in every review paper it's difficult to avoid the "laundry list paragraph of several different papers" trope, but minimizing this to whatever extent possible is ideal. Section 4.3.3 sticks out as a particular "laundry list" paragraph that outlines a dozen or so advance sin Transformer architectures without really unifying or describing the main shortcomings people are trying to work around, and the main trends in this research. There are also a lot of terms that are not fully explained and digestible to the average reader (e.g. CVAE, "drop-span embedding", etc)

## Overall

I don't think any of the weaknesses I described above are blockers to publication, and co-reviewers may not see as much issue with the framing as I do, and I am happy to be overriden. However in my opinion there is some additional work that would elevate this paper beyond the typical review paper in providing a more nuanced and unified view of the various new advances that have defined modern (2020+) RL research, and how precisely the Transformer architecture plays a role in all of this.

---

### Public Comment · ~Jack_Parker-Holder1 · 2023-07-07
**Missing Reference**

This paper uses transformers to train a generalist agent that can adapt in context to unseen environments:

Team et al. [Human-Timescale Adaptation in an Open-Ended Task Space](https://arxiv.org/abs/2301.07608), *ICML 2023*

As far as I know this is one of the largest-scale transformer-based agents trained with online RL.

---

> ### Author Response · Authors · 2023-07-13
> **Thanks for the reference**
>
> Thank you for your suggestion on additional references, and we will add this paper to the revised version of our survey.
>
> More specifically, this work demonstrates transformers' capability of improving adaptation and scalability in meta-RL problems, and we would like to include it in Section 4.4.1: Generalize to multiple tasks.

---

### Decision · Action_Editors · 2023-09-15

**Recommendation:** Accept with minor revision

**Comment:**

The reviewers all left a number of requested changes to particular points, and the framing in general. I see the authors have clarified in the responses, and I would expect them to update the paper accordingly before the final version is submitted.

**Audience:**

I think anyone interested in sequential decision making problems will be interested in this paper, which is a decent sized crowd.

**Claims And Evidence:**

This paper is a survey, so this question doesn't necessarily fully apply. The reviewers are satisfied that the coverage of relevant work, while perfect (an impossible task) is quite decent and of the standard expected for a survey at TMLR.

---

> ### Author Response · Authors · 2023-09-18
> **Thank you for your response!**
>
> Thank you very much for your appreciation of our work and for taking the time and effort to engage in discussions. We have updated the camera-ready version of our paper based on the reviewers' requested changes and our response. Please contact us if you have any questions.